# Experimental Study on Fluidization Behaviors of Wet Rice Threshed Materials with Hot Airflow

Tao Zhang [1,2], Yaoming Li [1,2,*], Lizhang Xu [1,2], Yanbin Liu [1,2], Kuizhou Ji [1,2] and Sheng Jiang [1,2]

1   Key Laboratory of Modern Agricultural Equipment and Technology, Jiangsu University, Zhenjiang 212013, China; jsdxzt@foxmail.com (T.Z.); justxlz@ujs.edu.cn (L.X.); 2111916005@stmail.ujs.edu.cn (Y.L.); jkz_jkzujs@163.com (K.J.); 2221916018@stmail.ujs.edu.cn (S.J.)
2   School of Agricultural Engineering, Jiangsu University, Zhenjiang 212013, China
*   Correspondence: ymli@ujs.edu.cn

**Abstract:** Among food crops, rice has the largest planting area, the highest yield per unit area and the largest total yield in China. However, cleaning performance is reduced by the high moisture content of rice during the harvesting process. In order to decrease the adhesion among wet rice threshed mixtures and improve the cleaning performance, the method of hot airflow cleaning was proposed. Firstly, the fluidization characteristics of wet sticky rice under the action of hot air were compared with those of dry particles. In this work, the minimum suspension velocity and the fastest suspension time were used and quantified to characterize the fluidization characteristics. It was found that the minimum suspension velocity and the fastest suspension time of the wet rice threshed mixture are both higher than those of dry particles due to the liquid bridge force. Moreover, the dispersion degree of the wet rice threshed mixture can be improved by the hot airflow due to the decrease in the surface water content of impurities. Secondly, the influence of the temperature and vibration frequency in the air-and-screen cleaning device on the dispersion characteristics of the wet rice threshed mixture were investigated. The accumulation mass was measured to quantify the dispersion degree. It was found that the increase in vibration frequency has little effect on the dispersion of the wet rice threshed mixture. The accumulation mass on the front of the sieve decreases slightly with the increase in the gas temperature range from ambient temperature to 30 °C. Then, the dispersion degree increases rapidly when the temperature exceeds 40 °C. The dispersion effect is the best when the temperature is 50 °C and the vibration frequency is 5 Hz. These results provide a basis for the cleaning of the wet rice threshed mixture in a combine harvester.

**Keywords:** wet rice; adhesion; hot airflow; dispersion characteristic; minimum suspension speed; surface water content

## 1. Introduction

Rice is the largest grain crop in China due to its high starch content [1–3]. During the rice harvest season, the climate is changeable and the weather is humid, so the moisture content of rice is too high when harvesting [4,5]. The adhesion between wet rice threshed materials occurs easily, resulting in the blockage of the cleaning screen, which seriously affects the rice quality and the working reliability of the combine harvester [6]. Hence, the air-and-screen cleaning device has been widely used in medium and large grain combine harvesters due to its high efficiency and strong adaptability [7–10]. To reduce the adhesion between the wet threshed mixture and the cleaning sieve, various methods, such as surface modification and introduction of a bionic screen, have been investigated. Ma et al. [11–14] applied the bionic non-smooth surface to the vibration screen of a rape harvester, and the test proved that the bionic non-smooth screen surface has good stability and use effect; however, the processing technology of the non-smooth screen is complex and the production cost is relatively high. Chen et al. [15] added a polytetrafluoroethylene coating to the surface of the vibrating screen to achieve the hydrophobic effect. Xu et al. [16] found

that the wettability of the rape cleaning screen surface can be reduced by a laser texture. Recently, a method to reduce the adhesion of wet sticky rice materials and the metal jitter plate by interface heating was proposed by Chen et al. [17]. The results indicated that the interface heating effectively decreased the adhesion rate between wet sticky rice materials and the jitter plate.

Before the working parameters can be determined, the influence of the working parameters on fluidization characteristics of the wet threshed mixture needs to be investigated and understood. However, only the fluidization state of the dry grain threshed mixture has been experimentally and numerically studied by a number of scholars. Chen et al. [18] studied the distribution of grain on the non-uniform air-flow cleaning unit by experiment. Li et al. [19] developed a CFD-DEM method to simulate the screening process of the air-and-screen cleaning device. The effect of the inlet airflow velocity was studied and analyzed in terms of grains and short straws' longitudinal velocity and vertical height, and cleaning loss. Bai Gali of Inner Mongolia Agricultural University [20] used a high-speed camera system to continuously collect the images of the flow field in the vertical pipe, measured the suspension velocity of several common grain particles, analyzed the influence of various factors on the suspension velocity of particle groups, and concluded that the suspension velocity of grain particles decreases with the increase in loading, and increases with the increase in the equivalent ball diameter. Hou et al. [21] studied the effect of moisture content of millet on floating speed and found that the floating speed of the grain, ear petal, stem and leaf of foxtail millet decreased with the decrease in moisture content. On the basis of the discussion of paddy cleaning mechanisms, the influence of the frequency and pulse width rate on separation was investigated by Chen et al. [22].

The capillarity of water film is an important cause of the formation of the adhesion interface, and the adhesion interface model between wet rice and the steel plate was developed by Chen et al. [17]. The fluidization characteristics of wet and dry rice threshed mixture were found to be apparently different. It was found that the understanding of fluidization characteristics of wet particles has been mainly advanced in chemical and medicine fields [23–26]. In order to reduce the agglomeration of the wet rice threshed mixture during the cleaning process, the hot airflow cleaning method was proposed. The fluidization characteristics of wet rice particles in a drying fluidized bed were first studied. Then, the dispersion characteristics of the wet rice threshed mixture were studied in the air-and-screen cleaning unit. This was helpful to reveal the dispersion mechanism of wet threshed materials and provide a theoretical basis for improving the performance of rice combine harvesters.

The objective of this study was to examine the effect of the airflow temperature on the fluidization state and dispersion characteristics in the drying fluidized bed. In addition, influences of the working parameters of cleaning device on the dispersion characteristic are also discussed in this paper.

## 2. Materials and Methods

### 2.1. Threshed Rice Mixture Samples

The test material was the indica rice planted in Miluo City, Hunan Province, China. After manual harvesting, a threshed rice mixture (see Figure 1) was obtained using the threshing device developed by Xu et al. [27].

The weight percentage of each component of rice after threshing was measured after sieving. The proportions of different components were 93.1% for rice grains, 0.7% for immature grains, 4.3% for short stems and 1.9% for light impurities. Immature grains are difficult to distinguish from grains in suspension, and their mass content is much lower than that of other materials, so they were not considered in this experiment. Each component of the threshed rice mixture was characterized by its density, equivalent diameter and moisture content. The physical properties of the rice threshed mixture were measured according to GB/T 24896-2010 and the literature [28]. The initial average values of the moisture content of rice grains, short stems and light impurities were 27.4%, 79.2% and

48.1%, respectively, and these materials were defined as wet particles, as shown in Table 1. Then, they were dried in the oven at 105 °C for 6 h and they were defined as dry particles [5]. Subsequently, these samples were kept in cold storage at 4–6 °C until further analysis.

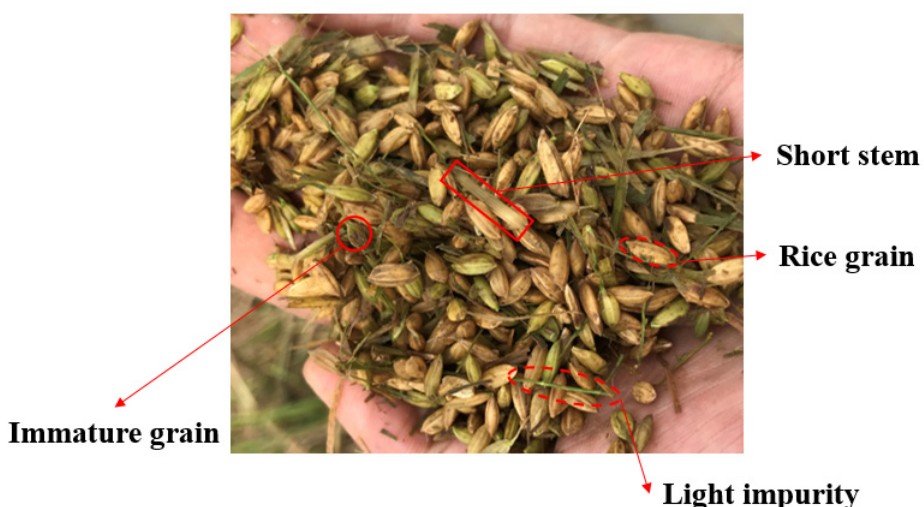

**Figure 1.** Wet rice threshed mixture.

**Table 1.** Physical properties of rice threshed mixture.

| Materials | Moist Content (%) | Length (mm) | Density (kg·m$^{-3}$) | Diameter (mm) |
|---|---|---|---|---|
| Rice grains | 15.3 [a]/27.4 [b] | 9.2 | 1085 | - |
| Short stems | 40.6 [a]/79.2 [b] | 70.5 | 650 | 2.1–3.9 |
| Light impurities | 16.9 [a]/48.1 [b] | 48.9 | 220 | 0.3–0.7 |

Note: [a] denotes dry particles; [b] denotes wet particles.

*2.2. Experimental*

2.2.1. Fluidized Bed Setup

A fluidized bed is commonly used to measure the suspension velocity of agricultural materials, such as wheat and cotton seed [29,30]. In this work, a drying fluidized bed was equipped to investigate the influence of hot airflow on the fluidization characteristics of the wet rice threshed mixture, as shown in Figure 2. A conical duct made of high light transmitting Plexiglas was used for observation. The height, bottom diameter and cone angle of the conical duct was 1000 mm, 140 mm and 6°, respectively. An asbestos net was located at the upper part of the steady flow cylinder to lay up rice particles and to ensure a good fluidization in the bed. Air was supplied by a centrifugal fan driven by an A.C. motor. The gas was able to reach the desired temperature by means of an electrical heater (200 kW) controlled by an A.C. contactor. Two ports were located in the side wall in the vertical direction of the column for pressure and temperature measurements. One of the ports was located at the bottom of the conical duct, and the other port was located 400 mm above the bottom port. The gas flow rate was changed gradually with a step of 18.3 m$^3$/h. The inlet airflow velocity was measured by an intelligent anemometer (LB-FS80). Pressure readings were taken with a differential pressure transducer (LFM11, ±1.0%FS, from China). Furthermore, the airflow temperature of the fan outlet was measured by the K Type Thermometer (TES-1310, Taipei, Taiwan) with an accuracy of ±0.1 °C.

Digital images were recorded with a high-speed camera (type of i-speed 3, OLYMPUS, Tokyo, Japan) equipped with a 12.5 mm lens. The frame of the camera was 150,000 fps. The recorded images consisted of 1280 × 1024 pixels. The running time required to reach steady state conditions was determined by conducting a series of experiments over gas flow velocities ranging between 0 and 12 m/s.

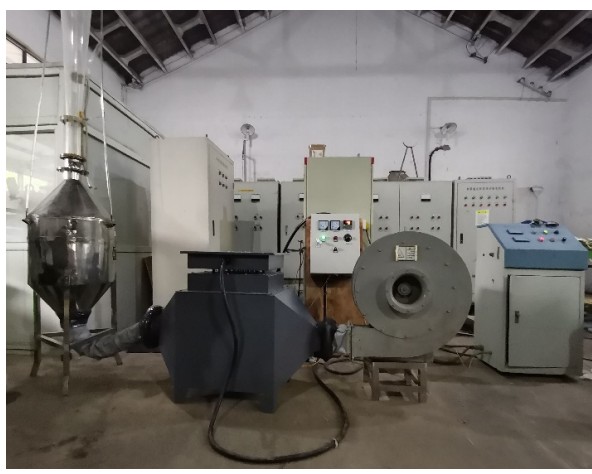
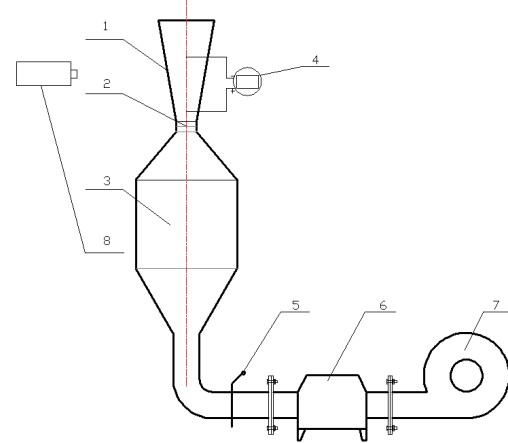

**Figure 2.** Scheme of the drying fluidized bed. 1. observed pipe; 2. asbestos net; 3. steady flow cylinder; 4. measuring port; 5. damper; 6. heating; 7. centrifugal fan; 8. high-speed camera.

2.2.2. Hot Airflow Cleaning Device

The cleaning unit was designed based on the previous study by Li [31], as shown in Figure 3, and the length and width of the cleaning sieve was 1200 × 960 mm. The outlet of heater was connected with the inlet of a cross-flow fan through a high-temperature resistant hose. The airflow temperature was adjusted by a heater (rated power 200 kW) and three thermometers (type of TM-902C, China) were used to measure the airflow temperature of the upper and lower fan outlets (test points 1, 2 and 3). The airflow temperature refers to the average value of the three test points.

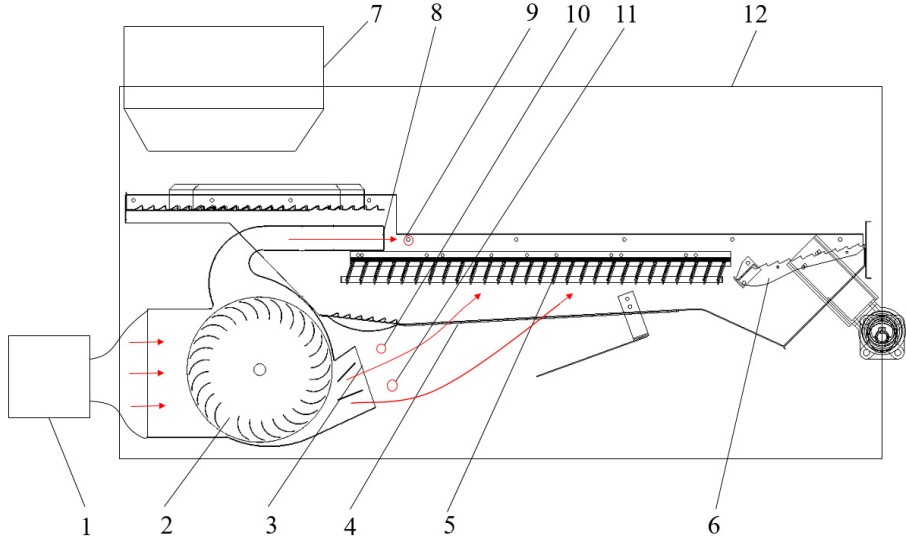

**Figure 3.** Sketch diagram of hot airflow. Cleaning test: 1. heater; 2. cross-flow fan; 3. lower gas outlet; 4. woven screen; 5. louvre screen; 6. tail screen; 7. hopper; 8. upper air outlet; 9. test point 1; 10. test point 2; 11. test point 3; 12. cleaning room.

*2.3. Experimental Procedure*

2.3.1. Fluidization Test

Two commonly used methods of measuring the terminal velocity experimentally are the drop and suspension methods [32]. The suspension method allows a particle to be suspended in a vertical duct by blowing air in a duct and measuring the air speed at the moment when the particle is suspended [33]. Study of fluidization characteristics was undertaken by investigating the effect of various airflow temperatures (12, 30, 40, 50 and 60 °C) on the minimum suspension velocity and fastest suspension time with the latter

method. Wet rice mixtures with a given composition (14 g rice grains, 4 g short stems, 6 g light impurities) were first loaded into the reactors. Then, the process gas velocity was measured through the bottom test port as the fan speed increased. The gas velocity and time when the rice changed from static to completely suspended in the drying fluidized bed were determined as the minimum suspension velocity and the fastest suspension time, respectively. In order to maintain a constant mass of material in the bed, the gas was turned off abruptly after the rice grains were completely suspended. Each experiment was conducted three times under the same conditions.

The pressure drop across the bed is an important parameter, characterizing the flow behavior in a fluidized bed [34]. In order to identify the influence of flow parameters on the dispersion characteristics of the wet threshed mixture, the dispersion degree of the wet rice threshed mixture was determined by analysis of the pressure drop across the tube (see Figure 2). The pressure drop was measured at different airflow temperatures at the operation time, t, of 8 s.

The agglomerates of wet rice grains, short stems and light impurities were related to the adhesion force caused by the surface water content of the mixtures [17]; hence, the surface water content of wet rice threshed mixtures against airflow temperature for different airflow velocities was measured according to the oven method (GB/T211-2007). The surface moisture content $\omega_s$ (wet base) was calculated by:

$$\omega_s = \frac{m_2 - m_1}{m_2} \times 100\% \tag{1}$$

where $m_2$ is the initial mass of wet particles (g); $m_1$ is the mass (g) after drying at 30 °C in the oven for 8 h.

### 2.3.2. Hot Airflow Cleaning Test

The wet rice threshed mixture was configured as follows: The fresh rice harvested from the field was placed into a drying box and dried at a constant temperature of 110 °C for 24 h to constant weight; then, the appropriate amount of water was poured into the dry mixture using a measuring cylinder of 10 mL accuracy, and the mixture was evenly mixed. The mixture was then sealed and placed in the refrigerator (0~4 °C) for 3 days. In this work, the wet threshed rice mixture was released from the hopper above the oscillating plate, the feed mass of the threshed rice mixture was 4 kg/s and the entire cleaning process time was 30 s. Each test under different conditions was repeated three times and the average value was applied. Operation parameters are presented in Table 2.

**Table 2.** Experimental parameters.

| Operation Parameters | Value |
|---|---|
| Heating temperature (°C) | 15, 30, 40, 50 |
| Vibration frequency (Hz) | 3, 4, 5, 6 |
| Vibration amplitude (mm) | 30 |
| Fan speed (r/min) | 1000 |

### 2.4. Statistical Calculations

Statistics in this article were calculated by the data fitting function of Orgin8.6 (Origin-Lab, Northampton, MA, USA).

## 3. Results and Discussion

### 3.1. Fluidization Characteristics of Wet and Dry Rice Threshed Mixture

3.1.1. Minimum Suspension Velocity and Fastest Suspension Time

The minimum suspension velocity is one of the most important hydrodynamic parameters that describes the gas–solid flow characteristics in a fluidized bed [35]. Figure 4 presents the effect of airflow temperature on the minimum suspension velocity $v_{mf}$ of dry

and wet rice threshed mixtures within 30 s. It can be seen that the minimum suspension velocity $v_{mf}$ of the dry threshed mixture is less than 6.5 m/s for all temperature levels, and was independent of the inlet gas temperature. For the wet rice threshed mixture, the average value of $v_{mf}$ is higher than 7.5 m/s at ambient temperature, which indicates a larger kinetic energy is needed for suspension. This is because the high moisture content will introduce the liquid bridge force between wet granules. The drag force needed for the wet rice threshed mixture to attain suspension in gas phase would be larger than that for dry particles. Moreover, the large porosity of the wet rice threshed mixture may also cause an increase in the minimum suspension velocity $v_{mf}$. From Figure 4, the minimum suspension speed $v_{mf}$ is almost constant when the temperature is less than 30 °C, and then decreases as the input airflow temperature increases. A possible reason for this is that the liquid bridge force in the packed bed decreases when the gas temperature increases from 30 to 40 °C because the liquid bridge force between wet particles decreases with the decrease in the surface tension of the liquid, which is dependent on temperature [35]. Furthermore, the liquid occupying the voids among wet particles would evaporate with the further increase in gas temperature. This would result in a smaller resistance to gas flow. When the airflow temperature exceeds 50 °C, the minimum suspension speed begins to stabilize. The minimum suspension speed of the wet threshed mixture is only 0.25 m/s larger than that of the dry mixture when the airflow temperature is 60 °C. Experimental data of the $v_{mf}$ of the wet threshed mixture were statistically fitted to empirical curves, which were given by the following equation:

$$v_{mf} = 6.2 + \frac{1.4}{1 + (T/35)^{11.7}} \tag{2}$$

where $v_{mf}$ is the minimum fluidized velocity (m/s); $T$ is the airflow temperature (°C). The adjusted R-square value of the polynomial is 0.88667, which indicates the polynomial can accurately express the mathematical relation between $v_{mf}$ and temperature $T$.

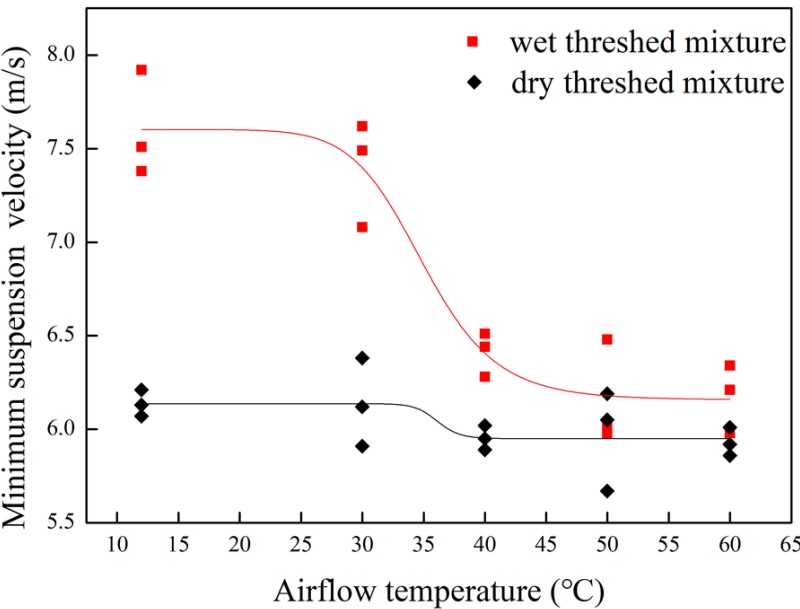

**Figure 4.** Minimum suspension velocity with temperature.

The time when most threshed mixture begin to be suspended in the test bed was defined as the fastest suspension time. Figure 5 shows the fastest suspension time at different airflow temperatures. A decrease in the fastest suspension time with increasing temperature was observed to occur for the wet threshed mixture. In comparison to the dry rice threshed mixture, the fastest suspension time of the wet rice threshed mixture was 27.3 s at the airflow temperature of 12 °C, which was 25 times longer than that of

dry particles. Moreover, focusing on the wet particles, the sudden decline in the fastest suspension time can be observed at the airflow temperature of 50 °C. The reason for this is that the agglomerates were broken down, thus enabling the system to fluidize. The fastest suspension time of the wet rice threshed mixture was less than 2 s when the temperature rose to 60 °C, as illustrated in Figure 5, which was close to the case of dry particles. The results show that the wet rice threshed mixture can be suspended within a very short time after heating.

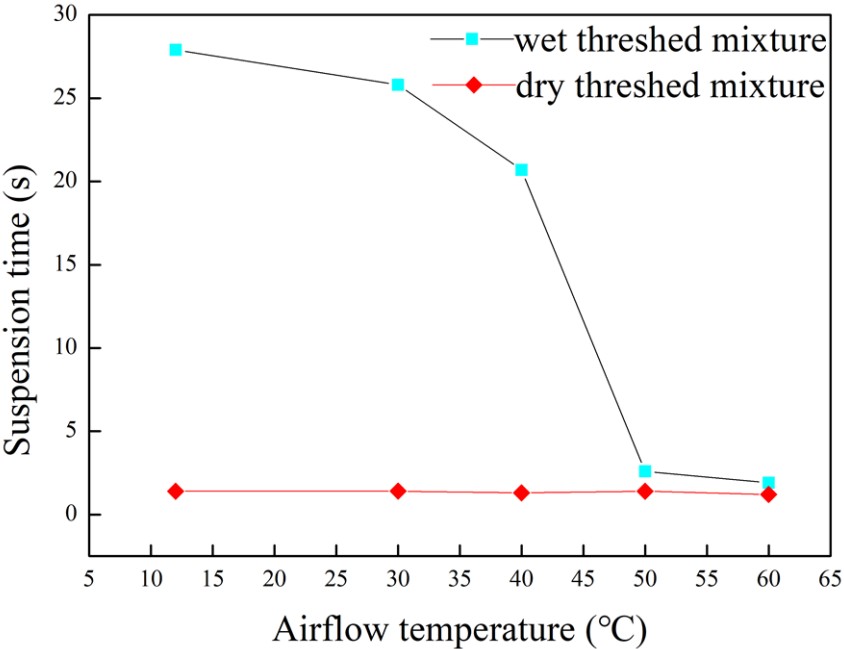

**Figure 5.** Suspension time with temperature.

3.1.2. Flow State

To better understand the effect of inlet gas temperature on fluidization behavior, some snapshots of the fluidization process were taken. According to the preliminary experiment of Figure 4, the minimum suspension velocities of the dry and wet threshed mixture were 6.14 and 7.60 m/s under ambient conditions, respectively. The inlet gas velocity $v$ was 7.60 m/s and working time t = 8 s. The fluidization process of dry and wet rice threshed mixtures under ambient conditions is illustrated in Figure 6a,b, respectively. Fluidization does not start until the inlet gas velocity is above the minimum suspension velocity of the rice threshed mixture. Thus, once enough air is injected into the drying fluidized bed, the dry and wet particle beds can be agitated by gas bubbles, and suspension of the threshed mixtures (including rice grains, short stems and light impurities) happens. For dry particles, it can be seen that the dry threshed mixture was completely suspended and scattered in the drying fluidized bed. In addition, these dry particles rotated and tumbled in the suspension device due to their irregular shape, and large gaps among them can be observed. For wet particles, however, the agglomerate made from the wet light impurities, rice grains and short stems were suspended in the upper position of the drying fluidized bed. As suspension continued, the agglomerate hardly changed. The reason for this is that the adhesion force among the wet rice threshed mixture is far larger than the drag force. In addition, the rough surface and large length–diameter ratio of the rice threshed mixture may also cause the agglomeration.

Figure 7 shows the flow state of the wet rice threshed mixture at different gas temperatures at $v$ = 7.60 m/s. Compared with 12 °C (see Figure 6b), the suspension behavior of the wet threshed mixture at 30 °C was hardly changed at t = 5 s, as shown in Figure 7a. A further increase in gas temperature caused the fluidization behavior to change, with the gas passing through horizontal channels. The lower half of the agglomerate was the first

to disperse and the short stems began to pass through the rice grain layer at the airflow temperature of 40 °C, as shown in Figure 7b. Then, a certain degree of dispersion occurred between rice grains and light impurities until the temperature was 50 °C (Figure 7c). At the same time, the flow pattern evolution of the wet threshed mixture in the test bed was similar to that of dry threshed mixture (see Figure 7a). According to Figure 7d, when the airflow temperature was 60 °C, all the rice grains and short stems were separated, and only a small number of light impurities were still reunited. This may be due to the intermolecular van der Waals force.

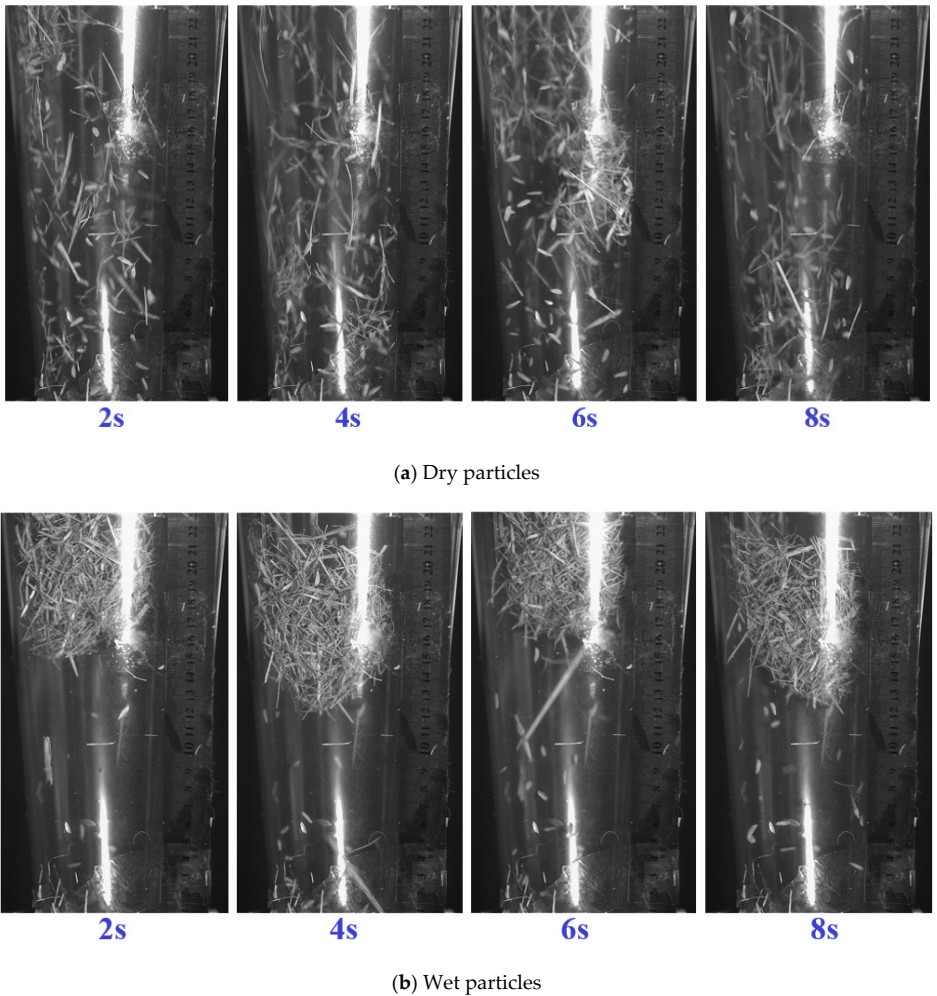

**Figure 6.** Flow state of rice threshed mixture ($v$ = 7.60 m/s and t = 8 s).

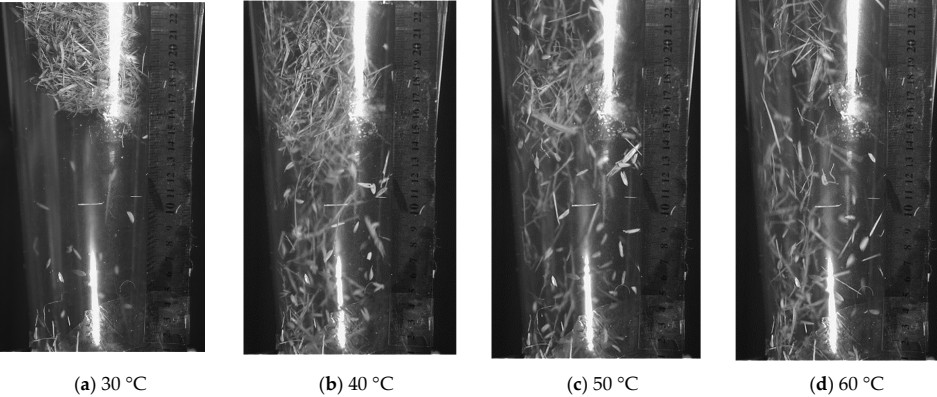

**Figure 7.** Flow state of wet threshed mixture at different airflow temperatures ($v$ = 7.60 m/s and t = 8 s).

### 3.1.3. Dispersion Degree of Wet Rice Threshed Mixture

The results reported in this work were obtained under steady-state operating conditions at t = 8 s. Figure 8 shows the curve of pressure drop versus gas temperature across the bed. Room temperature (12 °C) resulted in the largest pressure drop of 95.2 pa. When the inlet gas temperature was in the range of 12 to 40 °C, the pressure drop decreased slowly with increasing temperature. This indicates that the dispersion degree among wet particles varied little, which confirmed the phenomenon observed in Figure 7b. When the inlet gas temperature increased from 40 to 60 °C, the pressure drop reduced significantly and the dropping rate exceeded 80%. At the gas temperature of 60 °C, the pressure drop reached the minimum, i.e., 8.3 Pa in the present condition. The results of the above analysis show that heating can improve the dispersion degree of the wet threshed mixture in the fluidized bed. Experimental data of pressure drop versus airflow temperature were statistically fitted to empirical curves, given by the following equation:

$$\Delta P = 136.5 - 5.11T + 0.18T^2 - 0.002T^3 \tag{3}$$

where $\Delta P$ is the pressure drop (Pa); $T$ is the airflow temperature (°C). The adjusted R-square value of this polynomial is 0.97695, which indicates the polynomial can accurately express the mathematical relation between pressure drop $\Delta P$ and temperature $T$.

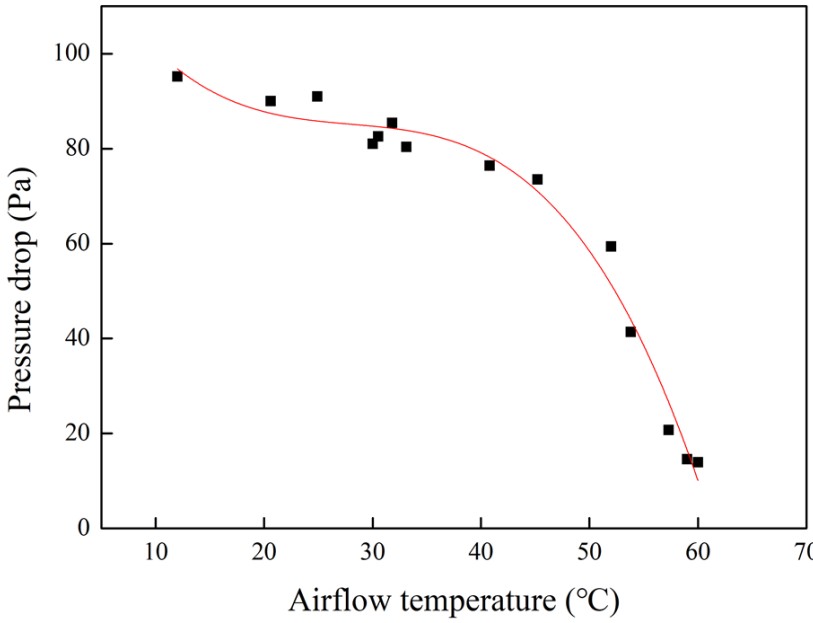

**Figure 8.** Pressure drop as a function of airflow temperature ($v$ = 7.60 m/s and t = 8 s).

### 3.1.4. Surface Water Content

As discussed above, the agglomerates of wet rice grains, short stems and light impurities were separated when the airflow temperature rose from ambient temperature to 60 °C. It was unknown whether this was related to the change in adhesion force caused by the surface water content of the mixtures. Figure 9 shows the surface moisture content of the wet threshed mixtures versus airflow temperature under steady state fluidization conditions at t = 8 s. A decrease in surface moisture with temperature can be observed for rice grains, short stems and light impurities. The dropping trend in the surface moisture content of light impurities is consistent with that in pressure. This indicates that the decrease in surface water content of light impurities is the main reason for the increase in dispersion degree. The moisture content of the surface of rice grains and short stems was 7.8% and 14.7% without heating, respectively, and hardly changed as the inlet gas temperature increased in the range of 12 to 40 °C, which also reveals the quite small dispersion degree of the wet rice threshed mixture seen from Figure 7a,b. Additionally, the surface moisture content of rice

grains and short stems was 3.4% and 7.9% when the airflow temperature was 50 °C, which clearly explains the reason for the good dispersion between wet rice grains and short stems observed in Figure 7c.

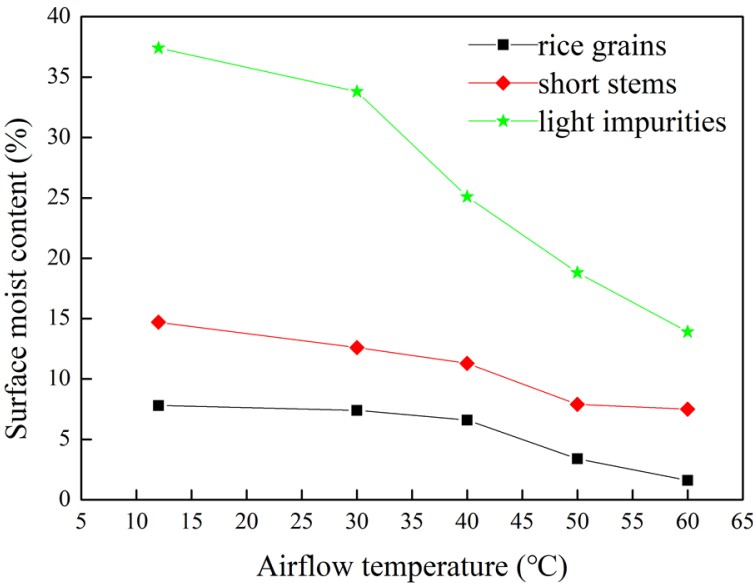

**Figure 9.** Curves of surface moisture content vs. temperature ($v$ = 7.60 m/s and t = 8 s).

### 3.2. Hot Airflow Cleaning Test

According to the above fluidization experiment, the dispersion degree of wet rice threshed mixture in the drying fluidized bed can be improved by hot airflow. However, the adhesion phenomenon among wet rice threshed mixtures usually arises under the vibration of the cleaning sieve of the combine harvester, and the agglomerates formed by the wet rice threshed mixture block the cleaning screen. Hence, the effect of hot airflow on the dispersion characteristics of the wet rice mixture in the air-and-screen cleaning device was studied.

Figure 10 presents a snapshot of the screen surface after the screening process finished. It can be seen that the wet threshed mixture entered the front of the sieve and accumulated there. Therefore, in order to quantitatively analyze the influence of cleaning parameters on the dispersion characteristics, the accumulation mass on the front half of cleaning sieve was measured after each test was completed. At the end of each test, the surface of the cleaning sieve was cleaned by towels to reduce experimental error.

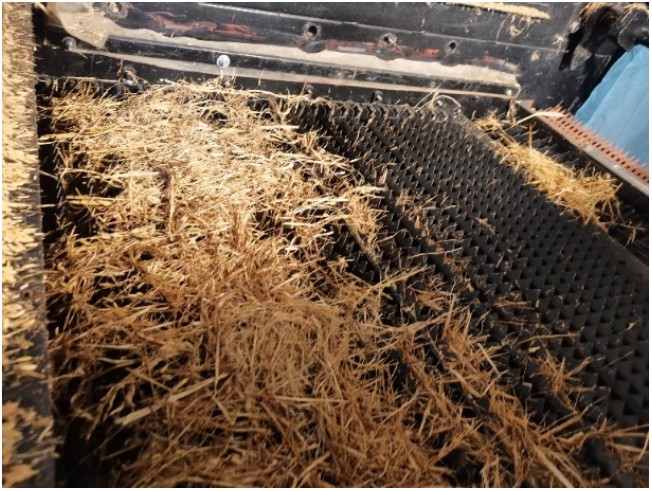

**Figure 10.** Snapshot of the cleaning screen surface.

The results of the accumulation mass are presented in Figure 11. It can be concluded that, for all vibration frequency levels, the accumulation mass of the wet threshed mixture on the cleaning sieve decreased with the increasing temperature. When the airflow temperature reached 50 °C, the accumulation mass was less than 250 g at 4 and 5 Hz. In addition, the accumulation mass differed little at various vibration frequencies when the airflow temperature was lower than 40 °C. When the airflow temperature and vibration frequency were 50 °C and 5 Hz, respectively, the accumulation mass on the front of the cleaning sieve was the smallest. As a result, heating can effectively reduce the accumulation on the screen surface, so as to improve the dispersion degree of the wet rice threshed mixture. However, the accumulation mass did not decrease to zero after heating; this may be due to insufficient temperature or other cleaning parameters.

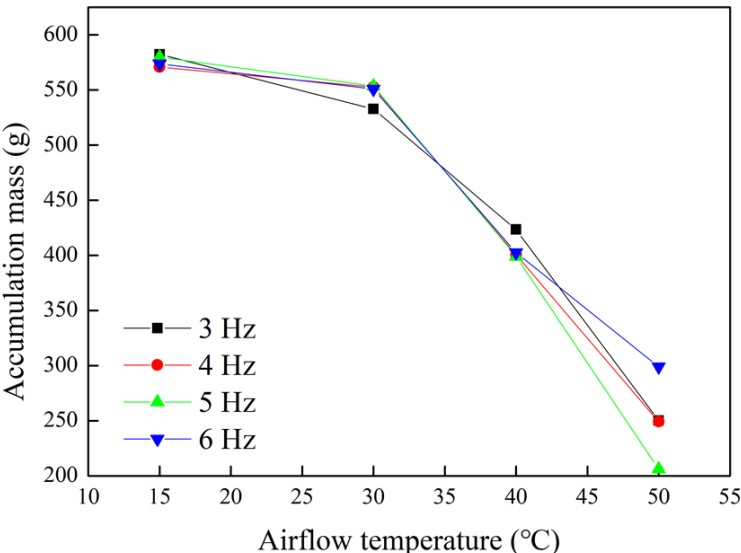

**Figure 11.** Effect of airflow temperature on accumulation mass.

## 4. Conclusions

Based on the results of the above study, the following main conclusions can be drawn:

1.  With the increase in temperature, the minimum suspension velocity and the fastest suspension time decreased gradually. The minimum suspension velocity of rice grains and short stems increased slightly with inlet airflow up to the critical temperature, equal to 40 °C, and then increased markedly with temperature. The fastest suspension time of the wet rice threshed mixture was less than 2 s when the temperature increased to 60 °C. According to the observed flow state of the wet rice threshed mixture, the agglomerations formed by rice grains, short stems and light impurities are difficult to break into discrete particles under the normal temperature airflow. The lower half of the agglomerate was the first to disperse and the short stems began to pass through the rice grain layer with the increasing temperature. When the gas temperature was about 60 °C, rice grains and impurities were effectively separated.

2.  The trends in the pressure drop and surface moisture content of light impurities were the same: both decreased gradually with the increase in inlet gas temperature. As the temperature increased to 60 °C, the surface moisture content of the wet particles decreased to a lesser extent, which explains the dispersion phenomenon in the fluidization process.

3.  The adhesion mass on the front of the screen surface can be reduced with the increase in the inlet airflow temperature. However, it is little affected by the vibration frequency. When the airflow temperature and vibration frequency were 50 °C and 5 Hz, respectively, the accumulation mass was less than 220 g. Another beneficial effect of increasing the inlet gas temperature is that the screen sieve was heated and the adhesion was reduced.

4.　In future work, more cleaning parameters, such as airflow velocity, louvre screen opening and vibration amplitude, will be considered to study the dispersion characteristics of the wet rice threshed mixture in the air-and-screen cleaning unit.

**Author Contributions:** In the research of this article, T.Z. mainly completed the writing and experimental part of the paper. Y.L. (Yaoming Li) and L.X. provided the guidance of scientific research funds and experimental scheme. K.J. and Y.L. (Yanbin Liu) and S.J. assisted me in completing the experimental research and helped me complete some papers. All authors have read and agreed to the published version of the manuscript.

**Funding:** This research was funded by National Natural Science Foundation of China under Grant grant number (51975257).

**Institutional Review Board Statement:** Not applicable.

**Informed Consent Statement:** Not applicable.

**Acknowledgments:** The authors gratefully acknowledge the National Natural Science Foundation of China.

**Conflicts of Interest:** We ensured that there was no conflict of interest.

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
