# Peer review of "Experimental Study on Fluidization Behaviors of Wet Rice Threshed Materials with Hot Airflow"

_agriculture, doi:10.3390/agriculture12050601_

Round 1

Reviewer 1 Report

The work deals with a very interesting issue related to the improvement of the rice grain harvesting process, which is a material characterized by high volatility at the stage of combine harvesting. Due to the existence of huge acreage of cultivation of this plant, even a slight modification of this process is worth attention, because it can bring very tangible benefits. Therefore, all work related to this topic is very important in terms of ensuring an adequate level of food production for people in the world. In terms of content, the text submitted to me does not raise any major objections, it only requires some sorting out.

And so, in my opinion, the aim of the research should be more emphasized, which is hidden in the last paragraph of the Introduction section. The text contained therein takes the form of a summary of what has been done in this paper, rather than what scientific problem will be solved in it.

In the Materials and Methods section, it is not stated at which air temperatures the experiment with fluidization of the rice mixture was conducted, and at which the pressure drop was determined. There is also no subsection on statistical calculations, although some formulas resulting from the statistical analyzes of the recorded results are presented later in the paper. The authors did not even provide the coefficient of determination of these equations, so it is not known how exactly they fit the experimental data. In the case of the caption of Figure 7, it is necessary to clarify what the case is considered, as it was presented in the caption of Figure 8. And here a small digression to the phrase given in line 215: temperature 12°C is not the room temperature level. The text between lines 230 and 235 should be moved to the Materials and Methods section. A reference to the procedure cited there should be found in the References section. Figure 10 shows no curves, only measurement points and there is no air velocity given there. The text between lines 254 and 274 together with figures 11 and 12 and table 2 should be moved to the previous section, as there is room for a description of the experimental procedure. Why were the same temperatures not used in the mass accumulation test as in the evaluation of the fluidization of the mixture?

In the Conclusion section, the first sentence is redundant. In turn, the last paragraph of this section should also be numbered, and the text should be arranged in the form of a program proposal, showing further activities in the implementation of research in this area.

The following minor issues should also be addressed during the revision process:

  • lines 4, 95, 96, 152, 173, 175, 180, 197, 198, 203, 205, 218, 236, 243, 246, 255, 259, 269 and 276 - no spaces,
  • lines 5-8 - arrange the given entries,
  • lines 10, 29 and 115 - remove extra space,
  • remove the bold font for the first words of the Abstract and Keywords section,
  • Keyword (should be Keywords) is too large font,
  • standardize the notation of the unit "°C" (with or without a space after the numerical value),
  • in section and subsection titles, subsequent words should start with a capital letter,
  • the first word of the title of the drawing and table and its headings should be written with a capital letter,
  • table 1 headings should be written in bold,
  • the measurement units from Table 1 should be entered in brackets,
  • figure 2 and 3 - dot in the wrong place,
  • lines 107, 113, 131, 161, 198, 213, 215, 267, 279 and 309 - no space between the number and the measurement unit,
  • for example, the measurement unit m3/h should be written as m3 h-1; correct other incorrectly written units,
  • line 116 - the name of the country should be written with a capital letter,
  • lines 121-123 - sentence written in a smaller font than the rest of the text,
  • explanations to Figures 4 and 11 should be provided after their titles,
  • lines 137 and 166 - no dot,
  • lines 144 and 160 - the sentence begins with a lowercase letter,
  • the formulas are written in enlarged type in relation to the rest of the text,
  • lines 162 and 168 - there is a comma instead of a dot,
  • line 172 - remove the comma,
  • line 173 - figure 3c is not included in the manuscript,
  • do not use the abbreviation "fig." throughout the work,
  • why in formula 3 the abbreviation "wet basic" is written with capital letters,
  • line 239 - what is "Tt" - no explanation of this abbreviation was given in the paper,
  • lines 245 and 304 - no unit °C,
  • figures 4 and 11 - use the same method of marking the drawn elements,
  • line 276 - no space from the table,
  • word Acknowledgment (should be Acknowledgments:) is written in all caps,
  • References section - there is no ordering of entries in this section; please refer to the guidelines in this journal's file template.

Author Response

Dear Reviewer:

Thank you for your comments. We have made the following modifications according to your suggestions. In addition, I have made other modifications to the manuscript, so we sincerely hope that you can review this manuscript again and make valuable suggestions. We believe this manuscript could provide readers with more valuable information.

Reviewer 2 Report

It is a technically focused paper, which perhaps falls into the scope of Agriculture scientific journal. The authors experimented with using heated air to clean the rice. I find this idea quite interesting with the potential use for cleaning rice during harvesting in humid conditions. I also consider the achieved results to be original, only its presentation should be better.

The introduction section is satisfactory and gives the reader a good overview of the individual parts of the problem to be solved. But it is not clear what the main aim of this article was. This should be clearly stated at the end of this section. Figure 1 is redundant.

In the Materials section, some information appears that belongs to the Results section and vice versa.

Now some detailed comment:

Materials

  1. 92-93: I would welcome some citation of the standard according to which the moisture content of the plant materials used was determined.
  2. 93-98, Table 1 – Results, not Materials.

Figure 3 is not necessary.

  1. 110 200 kW is really enough. Has an electric heater of such power really been used?
  2. 117: was measured, not can be. Please, use the past tense.

Fig. 4: The individual positions 1 to 8 should be in the caption of the image after its title, not in front of the caption of the image.

  1. 137 was conducted, not is conducted. Past tense please.

Results

  1. 164, Eq. (1) - for each equation, please complete the names of the variables and its units.
  2. 173: Fig. 3c - I did not find Fig. 3c anywhere in the text.

Fig. 5 - The measured values are interpolated by a curve. What is its equation? Is it equation (1)? If so, it should be indicated in the figure. The image should be understandable with the caption even without the surrounding text.

  1. 225 Eq. (2): variables and its units!
  2. 233 Eq. (3). How long was the material dried at 30 °C?
  3. 236: In this case, results are presented, not were.
  4. 242: 12 to 40 °C?
  5. 254 – 268, Fig. 11, Table 2: Materials, not Results.

Conclusion

  1. 290 – 291: First sentence is not needed.
  2. 306 – According to the airflow cleaning test – not needed.
  3. 309: …… temperature and vibration frequency was 50 °C and 5 Hz respectively, ….

Author Response

(The authors gave the same response as above.)

Round 2

Reviewer 2 Report

Article was improved. All my comments were addressed.